# Violence in Dating Relationships: Validation of the CADRI Questionnaire in a Young Adult Population

**DOI:** 10.3390/ijerph191711083

**Published:** 2022-09-04

**Authors:** Pedro José López-Barranco, Ismael Jiménez-Ruiz, César Leal-Costa, Elena Andina-Díaz, Ana I. López-Alonso, José Antonio Jiménez-Barbero

**Affiliations:** 1Faculty of Nursing, University of Murcia, 30120 Murcia, Spain; 2Department of Nursing and Physiotherapy, University of León, 24071 León, Spain

**Keywords:** young adults, gender, dating violence, perpetration, victimization

## Abstract

Dating violence in the young adult population is reaching alarming levels. However, the instruments used to measure it and their results are still heterogeneous. The main aim of this study was to validate the Conflict in Adolescent Dating Relationships Inventory questionnaire for a young adult Spanish university population aged between 19 and 25 years old, and to describe the types of violence perpetrated and suffered. Material and Methods. Observational, descriptive, cross-sectional study. A validity analysis was carried out through a confirmatory factor analysis. The relative frequencies for each type of violence and the Chi2 test for two dichotomous variables were used to describe the different types of violence perpetrated and suffered. Results: The sample consisted of 976 young adults aged 19–25 years old (M = 21.7 years. SD = 1.8). The confirmatory factor analysis had an adequate structure and a good fit to the model. The types of violence perpetrated and suffered were described according to the sex of the participants, with significant differences found for verbal-emotional violence and physical violence exerted. Conclusions: The confirmatory factor analysis allowed us to consider the application of the questionnaire to be correct for the study population.

## 1. Introduction

Violence in dating relationships (VDR) is defined as any violent psychological, physical, or sexual behavior, including bullying, directed towards a romantic partner [1,2,3]. Far from what has been traditionally believed about violence or abuse in relationships, a greater prevalence of abuse is not generally observed in adult relationships. Instead, a greater manifestation has been found in the younger population [4,5,6].

The VDR phenomenon is very broad and encompasses many types of abuse exerted and suffered. The types of abuse that are usually analyzed are: relational abuse (every act performed to socially humiliate or isolate the victim), verbal (every type of manipulation or aggression using language), psychological (every type of manipulation destined to humiliate and undermine the self-esteem of the victim, in public or private), physical (the use of physical force to inflict pain or suffering), and sexual (every non-desired sexual act, including comments that are sexual in nature) [7].

### 1.1. The Reach of VDR

Abuse in intimate partner relationships has reached alarming levels among adolescents and young people in the European context [8]. Abuse in the young adult and adolescent population is mainly bi-directional, in which both men and women are both victims and aggressors [9,10,11]. Although abuse in adolescents and adults is present at high levels, more severe acts and consequences are found in the young adult population [12,13].

The VDR outcomes have a considerable variability and heterogeneity, as a function of the evidence consulted. A systematic review carried out in 2017 in a young adult and adolescent population concluded that inflicted physical abuse ranged between 3.8% and 41.9% for women and between 7.7% and 40.3% for men. In the case of men, suffering physical abuse was found to be between 0.4% and 57.3%, and 1.2% and 41.2% for women; psychological abuse exerted was found to be between 4.2% and 97% for women and 4.3% and 95.3% for men; and psychological abuse suffered was found to be between 8.5% and 95.5% for men, and between 9.3% and 95.5% for women [14].

In 2021, a joint research study carried out by different member countries in the European Union noted that from their total adolescent sample aged between 13 and 16 years old, 34.1% of the girls and 26.7% of the boys were victims of VDR [15].

In Spain, these dynamics are similar. In 2019, a published study confirmed that in a sample of young Spanish individuals, 91.5% were identified as aggressors in their relationships, and 88.6% stated that they had been victims of these aggressions [16].

Although VDR is a bi-directional phenomenon in which both women and men exert and suffer different types of abuse, recent studies have shown that most of the actions of abuse identified and described as exerted by women tended not to be severe, or to be due to acts of self-defense against the aggressions exerted by men, with the women suffering highly brutal acts of abuse in greater numbers, such as sexual abuse, severe physical abuse, or even femicide [17]. On the other hand, women tended to exert slight physical abuse and psychological abuse in dating relationships to a greater degree [18,19,20].

The data on the greater victimization through severe forms of abuse against women seem to be incongruent with some studies that identify this group as the greatest perpetrators of different types of abuse as compared to men [6,16,17].

One of the main risk factors associated with suffering from VDR in later stages of life is to have suffered it in the early stages of life, including childhood or adolescence. Additionally, suffering VDR in the adolescent stages is directly associated with a greater perpetration and victimization of more severe types of abuse in adulthood or young adulthood [21,22,23]. VDR has also been identified with an important risk factor associated with suffering from severe health problems such as depression, anxiety, and drug consumption, sexual practices without protection, suicide, and even homicide [24,25,26,27].

### 1.2. Different Ways of Measuring Perpetration of and Victimization from VDR

The existing discrepancies in the results on the perpetration of and victimization from VDR could be due to the lack of consensus on which methodologies and instruments are adequate for measuring this type of abuse [28,29,30]. Some of the most-utilized questionnaires in Spanish that address VDR are the *Conflict in Adolescent Dating Relationships Inventory (CADRI)*, the *Questionnaire on Abuse Between Romantic Partners (CUVINO)*, or the *Conflict Tactics Scale (CTS)* [31,32,33]. One of the strongest and most widely used and tested instruments for the identification of this type of violence is the CADRI [34]. Moreover, this questionnaire can measure VRD exerted and suffered, thus providing data about both victims and aggressors [34]. This questionnaire was originally constructed by Wolfe et al. (2001) for an English-speaking adolescent population aged between 14 and 16 years old. The original questionnaire comprised 35 dual-nature items, which included two subscales: abuse exerted and abuse suffered. The different types of abuse analyzed and identified as dimensions in each of the sub-scales were relational abuse, verbal-emotional abuse, and physical abuse, sexual abuse, and threats. Although the authors present different types of violence, they do not emphasize the level of severity of the violence perpetrated or suffered.

The authors of the original scale only presented the reliability results for the subscale and dimensions of abuse exerted, which obtained a Cronbach’s alpha of 0.51 for sexual abuse, 0.52 for relational abuse, 0.82 for verbal-emotional abuse, 0.66 for threats, and 0.83 for physical abuse, with the reliability of the sum of the types of abuse exerted being 0.83. Fernández-Fuertes et al. (2006) [35] performed the first validation and adaptation of the CADRI in the Spanish language for a sample of adolescents aged between 15 and 19 years old. The authors concluded that the questionnaire was adequately adapted for the dimensions of verbal-emotional abuse and physical abuse, both exerted and suffered, with acceptable reliability indices, obtaining a Cronbach’s alpha of 0.78 for verbal-emotional abuse exerted, and 0.79 for that suffered; and 0.73 and 0.76 for the dimension of physical abuse exerted and suffered, respectively. The rest of the dimensions, although they had an adequate structural validity with respect to the original questionnaire, did not obtain sufficiently good reliability results. The abuse exerted and suffered obtained a Cronbach’s alpha of 0.59 and 0.73, respectively, the index of reliability for the sexual abuse exerted and suffered dimension was 0.56 and 0.55, respectively, and for threats, the indices of reliability were 0.53 for threats perpetrated and 0.51 for those suffered. Afterwards, Muñoz and Bandera (2014) [36] performed a factorial analysis of the CADRI for an older population, including adolescents and young adults aged between 17 and 21 years old. The authors concluded that the instrument was adequately adapted to the verbal-emotional abuse, relational abuse, and physical abuse dimensions. Their analysis obtained alpha values of 0.81 and 0.75 for physical violence exerted and suffered, respectively; values of 0.80 and 0.80 for verbal-emotional violence exerted and suffered, respectively, and in the relational violence dimension, alpha values of 0.52 and 0.71 for violence exerted and suffered, respectively.

The dimension of sexual abuse exerted and suffered did not perform well in the analysis, with an inadequate goodness-of-fit to the model, a high error, and low internal consistency, and therefore, the authors did not recommend the use of this dimension. The reliability index for each of the subscales after eliminating the dimension of sexual abuse was 0.81 for the abuse exerted and 0.84 for the abuse suffered.

A study published in 2020 utilized this questionnaire with a Spanish adolescent population aged between 12 and 17 years old [37]. The researchers utilized the dimensions of relational, verbal-emotional, and physical abuse for the sub-scales of abuse exerted and suffered, with a total of 17 dual-nature items. One of the main results indicated that men exerted more VDR than women. However, other research studies showed that women perpetrated more abuse in the relationships [14,38].

### 1.3. The Present Study

Given the background on the different validations of this instrument in the Spanish context, and the contradictory results observed in their validations and types of abuse identified as a function of sex, it was deemed necessary to delve into the validation of this questionnaire and to analyze its behavior in the population of interest. The availability of an instrument such as CADRI, adapted and validated to the Spanish university context for a population aged between 19 and 25 years old, provided us with the opportunity to more precisely diagnose this type of phenomenon in the young adult population in order to then perform interventions centered on reducing its incidence.

The research questions in the present study were: Is CADRI an adequate instrument for a young adult Spanish university population (19–25 years old)? Are there differences in the VDR exerted and suffered as a function of sex? These questions were to be answered with the following objectives: to validate the CADRI questionnaire on a young adult Spanish university population (19–25 years old), and to describe the types of abuse exerted and suffered as a function of sex identified by the CADRI in dating relationships in a young Spanish university adult population (19–25 years old).

## 2. Materials and Methods

### 2.1. Design

Instrumental, observational, descriptive, and cross-sectional study.

### 2.2. Participants and Sample Size

Young Spanish adults enrolled in a university, aged between 19 and 25 years old [39]. To select the sample, a non-probabilistic, intentional sampling was performed among the students at the Universidad de Murcia. The sample was composed of *n* = 976 young adults aged between 19 and 25 years old (*M* = 21.7 years old. *SD* = 1.8). 

### 2.3. Study Variables

The study participants were asked about their gender, the gender of their current or former partner, and the academic area in which they were enrolled. The different types of abuse exerted and suffered in their current relationship or during their last relationship if they were not currently in one, were analyzed.

### 2.4. Instruments Utilized

The analysis of the types of abuse was performed through the use of the *Conflict in Adolescent Dating Relationships Inventory (CADRI)*, using the version validated in Spanish [35]. This CADRI version had a total of 34 items, divided into two sub-scales: abuse exerted and abuse suffered. The different types of abuse analyzed were: relational abuse exerted (items: 1, 8, 17), (e.g., I tried to isolate him or her from his/her group of friends), verbal-emotional abuse exerted (items: 2, 3, 5, 6, 7, 9, 10, 11, 13, 15), (e.g., I did something to make my partner jealous), physical abuse exerted (items: 4, 12, 14, 16), (e.g., I threw an object), relational abuse suffered (items: 18, 25, 34), (e.g., he or she tried to isolate me from my group of friends), verbal-emotional abuse suffered (items: 19, 20, 22, 23, 24, 26, 27, 28, 30, 32), (e.g., he or she did something to make me jealous), and physical abuse suffered (items: 21, 29, 31, 33), (e.g., he or she threw an object at me). The participants responded to each item by selecting one of the following response options: Never (this has not happened in the relationship), rarely (this has only happened on 1 or 2 occasions), sometimes (this has happened between 3 to 5 times), and frequently (this has happened 6 or more times). The values for each response were based on a Likert scale with four alternatives: never = 0, rarely = 1, sometimes = 2, frequently = 3.

### 2.5. Procedure

To collect the data, a survey was provided that included sociodemographic data and the CADRI questionnaire. It was completed voluntarily and anonymously through a virtual tool from the Universidad de Murcia with a link provided through the institutional email from the Universidad de Murcia. The collection of data was conducted during the home confinement restrictions due to the COVID-19 pandemic, from 31 March 2020 to 7 July 2020. Before completing the questionnaire, a brief introduction was presented about the subject and the terminology utilized to ensure that the participants understood the questions to be asked, as well as the contact information of the main author, and an acknowledgement for their participation in the study.

### 2.6. Data Analysis

The statistical analysis was performed with the IBM SPSS Statistics software for Windows, version 26 (IBM, Armonk, NY, USA), and Mplus 5 (Los Ángeles, CA, USA) [40,41].

A confirmatory factor analysis (CFA) was performed to analyze the degree to which the items of the scale shaped the initial scale [42]. For the CFA, the weighted least square mean and variance adjusted (WLSMV) method of estimation was utilized, as it is commonly used for categorical data. The goodness-of-fit of the data to the model was performed with the *χ*^2^/df test, the comparative fit index (CFI), the Tucker–Lewis index (TLI), and the root mean square error of approximation (RMSEA). An adequate fit was considered when *χ*^2^/df < 5, CFI > 0.90, TLI > 0.90 and RMSEA < 0.08 [43]. To analyze the normal distribution of the data, the Kolmogorov–Smirnov test was utilized. A descriptive analysis was performed using the sociodemographic data of interest. The different types of abuse exerted and suffered were dichotomized for all of the dimensions to create two groups for each of the dimensions. These were: (0 = never exerted or suffered abuse in their romantic relationships, and 1 = exerted or suffered abuse one or more times in their romantic relationship). To be able to dichotomize both types of responses, we used a 0 for individuals who never exerted or suffered abuse, and a 1 for those who had exerted or suffered abuse rarely, sometimes, or frequently [44]. To identify if differences existed in the abuse exerted or suffered as a function of sex, the Chi-square (*χ*^2^) test was utilized for two dichotomous categories; this analysis was performed for each of the dimensions of the questionnaire as a function of sex. The null hypothesis was that the proportion of abuse exerted and suffered between women and men was equal, that is, abuse was independent of sex. The alternative hypothesis was that differences existed in the type of abuse exerted and suffered as a function of sex.

### 2.7. Ethical Considerations

The national and international guidelines from the Declaration of Helsinki, the European Guidelines for Data Protection, and Law 3/2018 of Protection of Digital Data and Rights were followed [45,46]. The research study was approved by the Ethics Committee from the University of Murcia with registration number 2922/2020. The individuals who participated in the study were adults and offered their consent before completing the questionnaire.

## 3. Results

### 3.1. Internal Structure of the Scale-CFA

The indices for each of the subscales were 0.80 for the abuse exerted and 0.90 for the abuse suffered. The reliability of each of the dimensions was adequate, except for the relational abuse exerted, which obtained a low reliability. The results are shown in Table 1.

The model of the scale of six oblique factors with the 34 items, adapted to Spanish, was tested [35], obtaining CFA fit index values of *χ*^2^ = 1365.30 (df = 512; *p* < 0.001), *χ*^2^/df = 2.67, CFI = 0.963, TLI = 0.959 and RMSEA = 0.041 (90% CI = 0.039–0.044). The factorial loads oscillated between 0.294 for item 9 and 0.955 for item 8. As a function of the goodness-of-fit results, the data of the model were considered adequate. The factor loads of the items and of the standardized factors can be consulted in Figure 1.

In the CFA, the types of abuse were codified as: vvee (verbal-emotional abuse exerted), vfee (physical abuse exerted), vree (relational abuse exerted), vves (verbal-emotional abuse suffered), vfes (physical abuse suffered), and vres (relational abuse suffered).

### 3.2. Descriptive Results for the Sociodemographic Variables

The sample was composed of *n* = 976 young adults aged between 19 and 25 years old. Of these 976 young adults, 83.2% were women, while 16.8% were men. Most of the participants (46.9%) were enrolled in the field of health sciences. Table 2 shows the sociodemographic characteristics of the sample. The distribution of the data was not normal, as shown by the Kolmogorov–Smirnov value at *p* < 0.05, total violence perpetrated: K-S = 0.141 (df 976; *p* < 0.001), total violence suffered: K-S = 0.187 (df 976; *p* < 0.001).

### 3.3. Abuse Exerted as a Function of Sex

The overall results of the analysis, when summing up the three types of abuse detected by the CADRI questionnaire (relational abuse, verbal-emotional abuse, and physical abuse), showed that 89.1% of the participants indicated having exerted some type of abuse in their dating relationships, while 10.9% affirmed not having committed any abusive acts in their romantic relationships. The data obtained, itemized according to sex, and according to the subscales of abuse exerted (Table 3), showed that on the one hand, the women obtained the greatest percentages in all the dimensions of abuse exerted, although we must consider that the number of participating men (16.8%) was significantly smaller than the number of women (83.2%). On the other hand, it was observed that the percentage of abuse that was more commonly exerted was verbal-emotional, with 89.9% of the women and 83.5% of the men stating they had exerted it on some occasion in their romantic relationship. However, 86.9% of the women and 91.5% of the men indicated not having exerted any relational abuse, and 87.7% and 95.1% of the men recognized not having ever exerted physical abuse. The differences found as a function of sex were significantly different for verbal-emotional abuse exerted (*X*^2^ (95% CI) = 5.836, *p* = 0.016) and for physical abuse exerted (*X*^2^ (95% CI) = 7.669, *p* = 0.006).

### 3.4. Abuse Suffered as a Function of Sex

The combined data obtained from the CADRI, with respect to the types of abuse suffered, showed that from the total sample, 16% indicated not having any type of abuse in their relationships, and 84% indicated having suffered some type of abuse in their relationships. The data obtained for the abuse suffered, as a function of sex (Table 4), revealed that the percentage of men and women who were identified as victims was similar; 84.03% of the women indicated having suffered some type of abusive act from their partners, while the men were identified as victims in 84.05% of the cases. For the different types of abuse suffered, a larger percentage of the women reported having suffered relational abuse (24.3%) and verbal-emotional abuse (83.7%) than men (22.1% and 82.9%, respectively). The differences found as a function of sex were not significant for relational abuse (*X*^2^ (95% CI) = 0.352, *p* = 0.553) or for verbal-emotional abuse (*X*^2^ (95% CI) = 0.066, *p* = 0.797). A larger percentage of the men mentioned suffering physical abuse than women, although significant differences for this type of abuse were not found (*X*^2^ (95% CI) = 2.123. *p* = 0.145).

## 4. Discussion

The main objective of this study was to validate the CADRI instrument with a young adult Spanish university population aged between 19 and 25 years old, to posteriorly identify the abuse exerted and suffered in dating relationships for the studied sample. This was achieved by analyzing if differences existed as a function of sex for each of the subscales and dimensions of the questionnaire. The CFA results were good, and the use of CADRI was deemed statistically adequate for the population analyzed in the present study. The indices of reliability and structure of the questionnaire were stable and similar to those presented by the creators of the original scale. If we compare our results with those from the first Spanish validation and adaptation study for a population of adolescents aged from 15 to 19 years old, the results were in agreement, with similar reliability index and CFA results to those found for the subscales of abuse exerted and suffered, and the dimensions of verbal-emotional abuse and physical abuse. The authors of the first Spanish validation and adaptation of the CADRI study noted that the dimension of relational abuse exerted and suffered obtained limited results. The non-satisfactory results found for this dimension could be due to the scarce variability in the responses found for each of the items that comprised the relational abuse exerted, with most of the responses around 0 (which indicates that this type of abuse did not occur in the relationship). According to the authors of the original Spanish adaptation, the homogeneity of low-level responses could provoke an alteration in the reliability that could be obtained in a sample of young individuals diagnosed as an at-risk group [35,36].

The validation performed was very similar to that from 2014 with a sample of Spanish adolescents and young adults up to 21 years of age, with an adequate goodness-of-fit to the model obtained for the dimensions of verbal-emotional, physical, and relational abuse, and reliability indices that were in agreement for the sub-scales of abuse exerted; CFI = 0.94 and RMSEA = 0.025 (90% CI = 0.019–0.03) for the violence exerted; and CFI = 0.936 and RMSEA = 0.026 (90% CI = 0.022–0.031) for the violence suffered [36]. The analyses performed by international research studies utilizing the original questionnaire showed similar reliability data for abuse exerted and suffered, with differences only found for the dimension of relational abuse exerted [6,44].

As for the sample selected, the studies that focused on the analysis of VDR among the young population tended to be restricted to university or school spheres [47,48]. In the present study, the selection criteria were similar to the last validation of the CADRI for a sample of Spanish adolescents and young adults, broadening the age range to 25 years old [36].

The most common abuse exerted and suffered by the participants in the sample was verbal-emotional abuse, followed by relational abuse and physical abuse, with these results similar to those found by other authors who utilized the same instrument to identify the VDR [49].

As for the differences in the perpetration of abuse as a function of sex, the women obtained greater percentages as compared to the men, with significant differences observed for the abuse exerted in verbal-emotional abuse and physical abuse. These results, although striking, are in agreement with other, previous studies, which showed that women indicated having exerted more VDR than men with respect to physical and relational abuse [50,51,52,53,54]. However, many studies argue the opposite [16,37,55,56]. These discrepancies could be associated, aside from the difference in the number of men and women included in the study, to the limitations found when addressing this problem through self-administered questionnaires [57]. On the other hand, it has been described that women indicate every act of abuse, in its most basic form, as having been committed by them [58]. This may be due to a greater awareness and knowledge of VDR [17]. In the case of the men, the bias was marked by the social desirability bias, which strongly influenced them, so that they did not mention having exerted diverse types of abuse for fear of the social rejection that this implies [59]. 

The data provided by the European Agency, and many recent publications on the subject, point out that men tend to exert more severe types of abuse in dating relationships, such as severe physical abuse, sexual abuse, or even homicide [60,61,62]. In this sense, it is important to have in mind the existence of studies [12,63], that defend the idea that men have less tolerance to abuse in its many forms, greatly identifying themselves as victims, and to a lesser degree as perpetrators. Another of the aspects that could explain the data obtained is that questionnaires such as CADRI are not able to identify if the abusive acts committed by women were a response to a previous aggression [55], so it would interesting to use tools that are able to measure the causes of these acts of abuse in romantic relationships to be able to identify acts of self-defense [64]. This questionnaire does not discriminate between the severity of the aggressions either, equally defining lesser-degree abusive acts (i.e., insults or bad stares) and more severe violent acts (i.e., physical or sexual aggression) [65].

Given the above and the limitations of these questionnaires, and due to the social recognition found in the literature that men tend to exert more severe types of abuse [18], the researchers agree with the suggestion by Dosil et al. (2020) that these studies must be complemented with mixed methodologies.

As for the abuse suffered, the women were greatly identified as victims, with a higher proportion of victimization for verbal-emotional abuse and relational abuse [66], while the men indicated suffering from physical abuse to a greater degree [14,53]. Significant differences were not found for the subscale of abuse suffered. These results differed from those found by other authors, which indicated that women suffered significantly more VDR in all its dimensions [67,68]. One of the types of abuse that was homogenously identified by women in other studies was sexual abuse, exerted by men in a high percentage, and entirely suffered by women [69,70,71,72].

### Limitations

The sample had a scarce representation of men, which may lead to the lack of evidence about a series of results that could be obtained with a more even sample, as for the sex of the participants. One of the aspects that could limit the ability to generate results is only studying a university population, with the possibility of finding different results for other groups of young adults still up to question.

Another of the key elements that limits the results obtained is the type of instrument chosen to identify VDR. The CADRI provides us with information that could be easily altered, due to the social desirability bias in men, and the women’s manner of interpreting abuse; also, women tend to be more sincere in their answers, and are more committed to the participation solicited by researchers in the university sphere.

Lastly, it should be recognized that one of the elements that could bias the use of this questionnaire with a young adult population, although the results of the factorial analysis were adequate, was the use, for an adolescent population, of a language different from that in which the CADRI was originally created.

## 5. Conclusions

The conducted study achieved its main and secondary objectives. The use of the CADRI with a young Spanish university population between the ages of 19 and 25 years old was considered correct at the statistical level, with adequate reliability and validity except for the dimension of relational abuse exerted.

The results published in the last few years have been heterogeneous. It is thus the responsibility of researchers to conduct studies and interventions that address the VDR problem from a more holistic perspective, not only by analyzing data on its incidence, but also by trying to understand why this type of abuse begins, and why it is more common to observe it in the young population.

For future research studies, we thus propose a mixed quantitative and qualitative methodology. Aside from the questionnaires, different techniques should be utilized, such as structured or semi-structured interviews. Another possibility for health workers is to conduct these types of research studies at hospitals and out-of-hospital services, completing the quantitative and qualitative data with victim care reports, for a more efficient and earlier intervention against VDR. This could provide a more global view of the situation and make possible the development of more efficient interventions.

## Figures and Tables

**Figure 1 ijerph-19-11083-f001:**
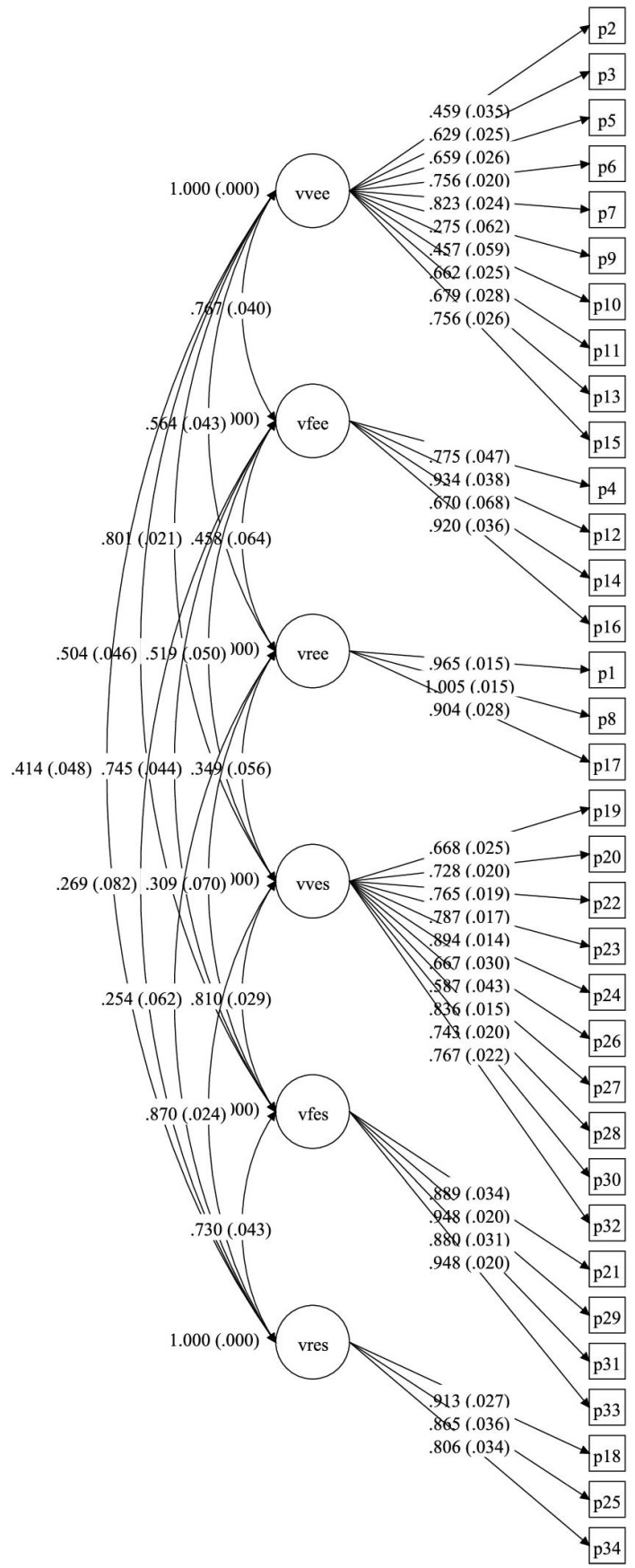
CADRI factor model.

**Table 1 ijerph-19-11083-t001:** Reliability indices for the CADRI in a young adult population.

	*Sub-Scale Abuse Exerted*		
**Types of abuse**	Relational abuse	Verbal-emotional abuse	Physical abuse
**Items**	1, 8, 17	2, 3, 5, 6, 7, 9, 10, 11, 13, 15	4, 12, 14, 16
**Cronbach’s alpha**	*0.83*	*0.79*	*0.69*
*Sub-scale abuse suffered*			
**Types of violence**	Relational abuse	Verbal-emotional abuse	Physical abuse
**Items**	18, 25, 34	19, 20, 22, 23, 24, 26, 27, 28, 30, 32	21, 29, 31, 33
**Cronbach’s alpha**	*0.66*	*0.88*	*0.84*

**Table 2 ijerph-19-11083-t002:** Sociodemographic characteristics.

Participant’s Sex	
Female	83.2% (*n* = 812)
Male	16.8% (*n* = 164)
**Field of study**	
Health sciences	46.9% (*n* = 459)
Social sciences	24.2% (*n* = 236)
Engineering	5.2% (*n* = 51)
Legal sciences	6.9% (*n* = 67)
Arts and humanities	15.7% (*n* = 153)
None of the above	0.9% (*n* = 9)
**Sex of your partner**	
Female	17.8% (*n* = 173)
Male	82.2% (*n* = 800)

**Table 3 ijerph-19-11083-t003:** Sub-scale of abuse exerted according to sex.

CADRI	Women		Men		
Exerted Abuse	Yes	No	Yes	No	*X* ^2^
*Relational abuse*	13.1% (*n* = 106)	86.9% (*n* = 706)	8.5% (*n* = 14)	91.5% (*n* = 150)	2.582 (*p* = 0.108)
*Verbal-emotional abuse*	89.9% (*n* = 731)	10.1% (*n* = 81)	83.5% (*n* = 137)	16.5% (*n* = 27)	5.836 (*p* = 0.016) *
*Physical abuse*	12.3% (*n* = 100)	87.7% (*n* = 712)	4.9% (*n* = 8)	95.1% (*n* = 156)	7.669 (*p* = 0.006) *

H0: There are no differences in abuse exerted or suffered as a function of sex *p* > 0.05. H1: There are differences in abuse exerted or suffered as a function of sex *p* < 0.05. * There are significant differences according to sex.

**Table 4 ijerph-19-11083-t004:** Subscale abuse suffered according to sex.

CADRI	Women		Men		*X* ^2^
Abuse Suffered	Yes	No	Yes	No	
*Relational abuse*	24.3% (*n* = 196)	75.7% (*n* = 612)	22.1% (*n* = 36)	77.9% (*n* = 127)	0.352 (*p* = 0.553)
*Verbal-emotional abuse*	83.7% (*n* = 680)	16.3% (*n* = 132)	82.9% (*n* = 136)	17.1% (*n* = 28)	0.066 (*p* = 0.797)
*Physical abuse*	11.2% (*n* = 91)	88.8% (*n* = 721)	15.2% (*n* = 25)	84.8% (*n* = 139)	2.123 (*p* = 0.145)

H0: There are no differences in abuse exerted or suffered as a function of sex *p* > 0.05. H1: There are differences in abuse exerted or suffered as a function of sex *p* < 0.05.

## Data Availability

The data are available upon email request to the corresponding authors.

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
