# Peer review of "Violence in Dating Relationships: Validation of the CADRI Questionnaire in a Young Adult Population"

_ijerph, 2022, doi:10.3390/ijerph191711083_

Round 1
Reviewer 1 Report
In their manuscript, the authors present data to validate the Conflict in Adolescent Dating Relationships Inventory (CADRI) in a large sample of Spanish adolescents. The factor structure is confirmed by means of confirmatory factor analysis. Furthermore, the results are compared across genders with a non-directional hypothesis to investigate whether male or female adolescents were the victims and perpetrators of different forms of abuse to a higher proportion.
Generally, I think that the study is important and well-conducted. The topic is a very important one for public health and the sample is large. However, I have a few major and minor points that should be addressed before publication. I hope that the authors will find my comments helpful.
1. The instrument should be described in more detail. Not having had contact with it in the past, the description of the questionnaire is not sufficient to understand it very well from the present article. It might be helpful to add the questions to the article as a supplement, but I do not know whether the authors have permission to do so, as the present study does not include the translation of the questionnaire, so the instrument has likely been published elsewhere. However, more item examples would be important. Also, I would like to better understand the severity of abuse covered by the instrument: what is an example of light abusive behavior, and does the instrument also cover extreme behaviors like rape and severe physical violence? Relatedly, what time frame does the instrument refer to? The authors state that the response alternatives refer to the frequency of occurrence of a behavior (this has never happened, it happened 2 or 3 times etc.), but I am unclear whether this refers to a specific period (e.g., during the past month), or whether this refers to the entire relationship the participant is currently in? If the latter is the case, how does the instrument control for the length of the current relationship? A 2-week relationship in which a partner was insulted twice seems more abusive than a 10-year relationship in which the partner was insulted twice.
2. The results seem rather scarce. The goodness-of-fit indices need to be reported at the least. Furthermore, since the validation of the questionnaire in the sample is the aim of the study, I would usually expect the factor structure to be also backed up by exploratory factor analysis or principal component analysis – what was the rationale of the authors for not performing these? Also, if male and female participants were compared, I would recommend a measurement invariance analysis. To be clear, I do not necessarily mean that the authors have to report all of these additional results, but I feel that the aim of the article (also stated in the title) and the methods used in the study need to match very well and a convincing case should be made that the methods used are adequate and sufficient to answer the research questions asked. However, I think that reporting fit indices is a must, and reporting measurement invariance is strongly suggested.
3. Relatedly, some details about the models are missing: were the factors standardized, and are the factor loadings reported standardized?
4. I have not really understood why the authors have chosen to dichotomize the questionnaire data and analyzed the probability of male and female adolescents belonging to a “no abuse” vs. “some abuse” sub-group. Could the answers in the questionnaire also be summed up to a sum score, and could the magnitude of this sum score then be compared across groups? Would this not be more informative, as it would also yield information about how severe the abuse suffered and exerted by the two groups was?
5. I think the Kolmogorov-Smirnov Test was mentioned in the Methods section, but the results were not reported.
6. The figure needs editing. The numbers are partly overlapping so that the reader can only see some of them. Also, it is not visible which covariance belongs to which path between factors. The figure should be self-explanatory with its figure caption; therefore, the authors should include a figure caption explaining the acronyms and all other information necessary to understand it.
7. In lines 90-106, the authors report Cronbach’s alpha coefficients from previous validation studies of their instrument, which range down to rather low values of below .60. I think this is worth discussing somewhere in the manuscript. This is consistent with the results from the present article, demonstrating that one of the scales has insufficient Cronbach’s alpha. Why does the instrument have such low reliability values? What about other forms of reliability, were these analyzed in previous studies or the current one? Would this not call for adaptation of the instrument?
8. The language throughout the article should undergo extensive editing. Particularly in the discussion, but also in the rest of the article, the sentences often seem vague, long, complex, and not very clearly phrased. In many places, I think to have also detected syntax errors.
There are also some minor points I would like mention:
· In l. 47-51, the authors report results from a review, but I was not quite sure what the percentages represent: participants reported to have exerted violence in X% of their relationships, and reported to have suffered violence in X%? Or X% of the participants reported violence? Or does this refer to different encounters, dates?
· In l. 66-68, the authors state that “The data on the greater victimization of severe forms of abuse against women seem to be incongruent with some studies that identify this group as the greatest perpetrators of different types of abuse as compared to the men.” However, this claim comes without references. It surprised me, because from the literature I know related to the context of violence and trauma, the perpetrators are more frequently male.
· L. 139: What do the authors mean by “instrumental study”?
· L. 141-142: What is meant by non-probabilistic, intentional sampling and what were the inclusion criteria?
· L. 170-171: The authors claim that the data were collected during the COVID-19 pandemic while the participants were in isolation. How could this have affected the relationship behavior and, consequently, the responses of the participants?
· Table 1: Should the second heading be “Sub-scale suffered abuse” or something like that?
Author Response
Reviewer 1:
1. The instrument should be described in more detail. Not having had contact with it in the past, the description of the questionnaire is not sufficient to understand it very well from the present article. It might be helpful to add the questions to the article as a supplement, but I do not know whether the authors have permission to do so, as the present study does not include the translation of the questionnaire, so the instrument has likely been published elsewhere. However, more item examples would be important. Also, I would like to better understand the severity of abuse covered by the instrument: what is an example of light abusive behavior, and does the instrument also cover extreme behaviors like rape and severe physical violence? Relatedly, what time frame does the instrument refer to? The authors state that the response alternatives refer to the frequency of occurrence of a behavior (this has never happened, it happened 2 or 3 times etc.), but I am unclear whether this refers to a specific period (e.g., during the past month), or whether this refers to the entire relationship the participant is currently in? If the latter is the case, how does the instrument control for the length of the current relationship? A 2-week relationship in which a partner was insulted twice seems more abusive than a 10-year relationship in which the partner was insulted twice.
Response:
Thank you very much for your evaluation, it has helped us to improve the manuscript.
The CADRI instrument analyzed does not consider what types of violence can be classified as more or less serious, as opposed to the M-CTS, which separates two types of physical violence: medium and serious. The CADRI dimensions validated in Spanish and used in the study analyzed verbal, relational and physical violence, without considering the severity of the latter.
We could understand relational violence such as: “I tried to separate him from his group of friends”, and verbal-emotional violence such as: “I spoke to him in a loud or offensive tone of voice” as mild forms of violence within CADRI. And within the dimension of physical violence, we find more serious acts such as "I kicked him, hit him or punched him." Although the authors do not define what types of violence can be considered serious or minor. (Page 2, section 1.2.), (page 4, section 2.4. lines 159-174, examples of items are shown for each type of violence analyzed)
The participants were asked about the frequency of abuse experienced or exerted during their current relationship or the last relationship if they were not currently in one. The questionnaire did not refer to the duration of the relationship, only being or having been involved in a romantic relationship.
Fernández-Fuertes, A. A. Evaluación de la violencia en las relaciones de pareja de los adolescentes. Validación del Conflict in Adolescent Dating Relationships Inventory (CADRI) - versión española. 2006, 6 (2), 21.
Muñoz, J. L. B.; Bandera, J. F. M. Análisis factorial de las puntuaciones del CADRI en adolescentes universitarios españoles. Universitas psychologica 2014, 13 (1), 175–186.
Muñoz-Rivas, M. J., Andreu Rodríguez, J. M., Graña Gómez, J. L., O’Leary, K. D., & González Lozano, M. P. (2007). Validación de la versión modificada de la Conflicts Tactics Scale (M-CTS) en población juvenil española. https://repositorio.ucjc.edu/handle/20.500.12020/681
2. The results seem rather scarce. The goodness-of-fit indices need to be reported at the least. Furthermore, since the validation of the questionnaire in the sample is the aim of the study, I would usually expect the factor structure to be also backed up by exploratory factor analysis or principal component analysis – what was the rationale of the authors for not performing these? Also, if male and female participants were compared, I would recommend a measurement invariance analysis. To be clear, I do not necessarily mean that the authors have to report all of these additional results, but I feel that the aim of the article (also stated in the title) and the methods used in the study need to match very well and a convincing case should be made that the methods used are adequate and sufficient to answer the research questions asked. However, I think that reporting fit indices is a must, and reporting measurement invariance is strongly suggested.
Response:
Thank you very much for your comment. The goodness-of-fit indices are presented below: CFA fit index values of χ2 = 1365.30 (df = 512; p < 0.001), χ2/df = 2.67, CFI = 0.963, TLI = 0.959 and RMSEA = 0.041 (90%CI = 0.039−0.044). These data have been included in the results section in section 3.1. Internal Structure of the Scale-CFA.
Following the recommendations, a CFA was carried out to verify the internal structure of the original questionnaire. Some authors proposed establishing the differentiation between both approaches, not according to their aim, but in the restrictions imposed. Thus, instead of considering the EFA and CFA as two qualitatively distinct categories, they should be considered as two ends of a continuum. Thus, the EFA (non-restrictive) imposes minimum restrictions for obtaining an initial factor solution, which can be transformed by applying different rotation criteria. And the CFA (restrictive) imposes much stronger constraints that allow a single solution to be tested, the fit of which can be evaluated using different goodness-of-fit indices (Lloret-Segura et al., 2014). For this reason, an EFA was not carried out since the objective was to analyze the final structure of the original questionnaire through a CFA.
Thus, a CFA was performed to analyze the degree in which the items of the scale shaped the initial questionnaire. For the CFA, the weighted least squares mean and variance adjusted (WLSMV) method of estimation was utilized, as it is commonly used for categorical data. The goodness-of-fit of the data to the model was performed with the χ2/df test, the comparative fit index (CFI), the Tucker-Lewis index (TLI), and the root mean square error of approximation (RMSEA). In the results section we have added the values of the goodness-of-fit indices (page 5, lines 215-216).
Lloret-Segura, S.; Ferreres-Traver, A.; Hernández-Baeza, A.; Tomás-Marco, I. Exploratory Item Factor Analysis: A practical guide revised and up-dated. Anales de Psicología / Annals of Psychology 2014, 30, 1151–1169, doi:10.6018/analesps.30.3.199361.
3. Relatedly, some details about the models are missing: were the factors standardized, and are the factor loadings reported standardized?
Response:
Thank you for your observation, the required information is included in section 3.1. Internal Structure of the Scale-CFA.
The factor loads of the items and of the standardized factors can be consulted in Figure 1. All were statistically significant (p<0.001).
4. I have not really understood why the authors have chosen to dichotomize the questionnaire data and analyzed the probability of male and female adolescents belonging to a “no abuse” vs. “some abuse” sub-group. Could the answers in the questionnaire also be summed up to a sum score, and could the magnitude of this sum score then be compared across groups? Would this not be more informative, as it would also yield information about how severe the abuse suffered and exerted by the two groups was?
Response: Initially, the authors evaluated dichotomizing the data to present the results in a more holistic way and for the reader to be able to more easily understand the rates of violence exerted and suffered. The dichotomization of the items allows us to indicate the existence or not of violence exerted or suffered. This simplifies the presentation of the data and allows us to provide specific information for the discussion, due to the heterogeneity reported in the studies that preceded the present one.
Dosil, M.; Jaureguizar, J.; Bernaras, E.; Sbicigo, J. B. Teen Dating Violence, Sexism, and Resilience: A Multivariate Analysis. IJERPH 2020, 17 (8), 2652. https://doi.org/10.3390/ijerph17082652.
5. I think the Kolmogorov-Smirnov Test was mentioned in the Methods section, but the results were not reported.
Response:
The results of the Kolmogorov-Smirnov test for each of the violence subscales are presented. (The data is presented in the results part page 7, section 3.2.)
Total violence perpetrated: K-S=0.141 (df 976; p<.001)
Total Violence suffered: K-S= 0.187 (df 976; p<.001)
6. The figure needs editing. The numbers are partly overlapping so that the reader can only see some of them. Also, it is not visible which covariance belongs to which path between factors. The figure should be self-explanatory with its figure caption; therefore, the authors should include a figure caption explaining the acronyms and all other information necessary to understand it.
Response: The recommendations of the reviewer were followed. The data is presented at the bottom of the figure to improve understanding. The size and quality of the image have been improved.
7. In lines 90-106, the authors report Cronbach’s alpha coefficients from previous validation studies of their instrument, which range down to rather low values of below .60. I think this is worth discussing somewhere in the manuscript. This is consistent with the results from the present article, demonstrating that one of the scales has insufficient Cronbach’s alpha. Why does the instrument have such low reliability values? What about other forms of reliability, were these analyzed in previous studies or the current one? Would this not call for adaptation of the instrument?
Response:
By reviewing the database, we obtained adequate reliability values for the dimension of relational violence exerted. This was due to an error on our part when performing the calculations without adjusting the response values appropriately. Thank you for the observation and we apologize for this mistake.
8. The language throughout the article should undergo extensive editing. Particularly in the discussion, but also in the rest of the article, the sentences often seem vague, long, complex, and not very clearly phrased. In many places, I think to have also detected syntax errors.
Response:
The text was sent to a professional editor for proofreading.
There are also some minor points I would like mention:
- In l. 47-51, the authors report results from a review, but I was not quite sure what the percentages represent: participants reported to have exerted violence in X% of their relationships, and reported to have suffered violence in X%? Or X% of the participants reported violence? Or does this refer to different encounters, dates?
Response:
The results refer to the different values of perpetration and victimization that were identified in a systematic review that integrated different studies. Modifications were made in the text to make this clearer.
- In l. 66-68, the authors state that “The data on the greater victimization of severe forms of abuse against women seem to be incongruent with some studies that identify this group as the greatest perpetrators of different types of abuse as compared to the men.” However, this claim comes without references. It surprised me, because from the literature I know related to the context of violence and trauma, the perpetrators are more frequently male.
Response:
In the study entitled "KNOWLEDGE, ATTITUDES AND PRACTICES OF SPANISH ADOLESCENTS ON PARTNER VIOLENCE", the authors present us with results for the perpetration and victimization of different types of violence. The authors identified a greater victimization by physical violence in men M=0.23 (SD 0.47) as compared to women M=0.13 (SD 0.32). These results are similar to those found in various studies that identified a greater perpetration of physical violence by women. These results are discussed in the Discussion section.
However, regarding the form of sexual violence, there was homogeneity in the greater victimization of women and greater perpetration by men, as with the most extreme form of violence "murder" in the context of the couple, with women being the main victims.
Taylor, B.; Mumford, E. A National Descriptive Portrait of Adolescent Relationship Abuse: Results From the National Survey on Teen Relationships and Intimate Violence. Journal of Interpersonal Violence 2016, 31 (6), 963–988. https://doi.org/10.1177/0886260514564070.
Dosil, M.; Jaureguizar, J.; Bermaras, E. Variables Related to Victimization and Perpetration of Dating Violence in Adolescents in Residential Care Settings. Spanish Journal of Psychology 2019, 22, 36. https://doi.org/10.1017/sjp.2019.35.
Bravo, M. del M. P.; Meseguer, C. B.; Llor, A. M. S.; Pina-Roche, F. Conocimientos, actitudes y prácticas de adolescentes españoles sobre la violencia de pareja. iQual. Revista de Género e Igualdad 2018, No. 1, 145–158. https://doi.org/10.6018/iQual.301161.
- 139: What do the authors mean by “instrumental study”?
Response:
When we talk about an instrumental design we refer to the study of a specific instrument, in this case the CADRI analysis, and its behavior in a specific population.
In this way, it is possible to validate an instrument for a specific population. In this case young adults between 19 and 25 years of age.
- 141-142: What is meant by non-probabilistic, intentional sampling and what were the inclusion criteria?
Response:
Non-probabilistic, intentional sampling: Non-probabilistic refers to when the sample elements are selected using non-random criteria. It is used when probability sampling is too expensive or not possible. Intentional for convenience refers to the selection of accessible individuals by the researcher when collecting data.
Inclusion criteria: Young Spanish university adults aged between 19 and 25 years old, who were in a current or former dating relationship
- 170-171: The authors claim that the data were collected during the COVID-19 pandemic while the participants were in isolation. How could this have affected the relationship behavior and, consequently, the responses of the participants?
Response:
The reference to the COVID-19 pandemic is to take into account the capacity for greater availability of time by the participants, thus achieving a higher response rate and a greater capacity for reflection on each item.
We believe that the period of confinement influenced the response rate for answering the questionnaire, thereby increasing the sample to a considerable number.
- Table 1: Should the second heading be “Sub-scale suffered abuse” or something like that?
Response:
Thanks for the review. Changes were made based on the reviewer's recommendation.

Reviewer 2 Report
Dear Authors,
It was a great pleasure to read your article. The topic is very actual and extremely important. And – what the most important – study design and research realization are very satisfactory.
The title is informative and accurately reflects the manuscript. The abstract is complete and stand-alone. It adequately reflects the content of the manuscript.
The Introduction provide sufficient theoretical background for the study. The theoretical framework is properly matched to the research problems being carried out. The introduction is structured logically. The study’s justification is stated clearly.
The research design answers the proposed research questions. The study’s methodology and the execution of the study are adequate. The important aspects of the methods are clearly described.
The results are clearly organized and presented. The analysis is adequately described. Tables and other visual material are clear and easy to interpret.
The structure of the Discussion is very clear. The interpretations are appropriate, supported by the results, and discussed with relevant literature and within the limits of the study.
I really have no doubt, that this article can be published in actual form, which is rare.
Kind regards
Author Response
Dear reviewer, thank you for the work done and the comments included in the process. We appreciate the very positive observations made about the text and especially the time you spent reading and analyzing the manuscript.
Reviewer 3 Report
Dear authors,
During the reading of your article, there were several moments when I felt lost with your objective.
The article does not justify the reason for presenting a new validation when there are already two (references 35 and 36), given that neither items nor dimensions were eliminated.
The methodology is inappropriate since it does not present concrete data on exploratory factor analysis (communalities, alphas, regressions, etc etc) and confirmatory factor analysis (CFI, GFI, TLI, RMESEA, SRMR)
Although the alpha value is very low for the "relational abuse" dimension (0.17) because they did not eliminate the items or do not justify the reasons for maintaining the dimension, or present convergent validity.
Comparative data between the present study and previous studies (CFI, GFI, TLI, RMESEA, SRMR) are also not shown.
The article does not justify the relevance of this study, so it is not important for readers.
Author Response
During the reading of your article, there were several moments when I felt lost with your objective.
Response: The study presented has two main objectives. On the one hand, to validate the questionnaire for a specific population group: young adults. This objective was proposed with the aim of having a validated questionnaire to measure the violence exercised and received in this population group. Previous validations did not include this population group. Once the validity of the questionnaire for this population group was confirmed, we were able to meet the second objective of the study: the measurement of violence exerted and received.
The article does not justify the reason for presenting a new validation when there are already two (references 35 and 36), given that neither items nor dimensions were eliminated.
Response: Thank you very much for your comments, they were very helpful. The validation of the CADRI allowed us to apply the questionnaire to a population between 19 and 25 years of age. This age range is broader than the other ones validated so far. This validation allowed us to use the instrument with a broader range of young adults.
The methodology is inappropriate since it does not present concrete data on exploratory factor analysis (communalities, alphas, regressions, etc etc) and confirmatory factor analysis (CFI, GFI, TLI, RMESEA, SRMR)
Response: Thank you very much for your comment. Following the latest recommendations, a CFA was carried out because the intention was to verify the internal structure of the original questionnaire. Some authors proposed establishing the differentiation between both approaches, not according to their aim, but on the restrictions imposed. Thus, instead of considering the EFA and CFA as two qualitatively distinct categories, they should be considered as two ends of a continuum. Thus, EFA (non-restrictive) imposes minimum restrictions for obtaining an initial factor solution, which can be transformed by applying different rotation criteria, and the CFA (restrictive) imposes much stronger constraints that allow a single solution to be tested, the fit of which can be evaluated using different goodness-of-fit indices (Lloret-Segura et al., 2014). For this reason, an EFA was not carried out, as the objective was to verify the final structure of the original questionnaire through a CFA.
Thus, a CFA was performed to analyze the degree in which the items of the scale shaped the initial questionnaire. For the CFA, the weighted least squares mean and variance adjusted (WLSMV) method of estimation was utilized, as it is commonly used for categorical data. The goodness-of-fit of the data to the model was performed with the χ2/df test, the comparative fit index (CFI), the Tucker-Lewis index (TLI), and the root mean square error of approximation (RMSEA). In the results section we have added the values of the goodness-of-fit indices (page 5, lines 215-216).
Lloret-Segura, S.; Ferreres-Traver, A.; Hernández-Baeza, A.; Tomás-Marco, I. Exploratory Item Factor Analysis: A practical guide revised and up-dated. Anales de Psicología / Annals of Psychology 2014, 30, 1151–1169, doi:10.6018/analesps.30.3.199361.
Although the alpha value is very low for the "relational abuse" dimension (0.17) because they did not eliminate the items or do not justify the reasons for maintaining the dimension, or present convergent validity.
Response:
After reviewing the original database and the results, we identified an error in the calculation of the alpha value for "exercised relational abuse". The changes in the document are specified (page 5, table 1).
Comparative data between the present study and previous studies (CFI, GFI, TLI, RMESEA, SRMR) are also not shown.
Response:
Thanks for the review. The data referring to the last validation carried out in Spanish of the CADRI questionnaire, are presented and integrated into the text to be able to compare them with those from our study. The changes can be found on page 9.
On page 3, the reliability values of the CADRI questionnaire in previous validation studies are integrated.
The article does not justify the relevance of this study, so it is not important for readers.
Response:
Thank you very much for the comment. The purpose of the study was to provide sufficient evidence to support the use of the CADRI in a sample of young adults up to 25 years of age, for whom an analysis of their behavior and structure had not been developed previously. In addition, we found it interesting to provide data on victimization and perpetration of violence in the young adult population, and to analyze what differences existed according to sex/gender. It was possible to demonstrate a reliable structure of the CADRI to be applied in the young adult population up to 25 years of age.